# Rise of the Machines: The Inevitable Evolution of Medicine and Medical Laboratories Intertwining with Artificial Intelligence—A Narrative Review

**DOI:** 10.3390/diagnostics11081399

**Published:** 2021-08-02

**Authors:** Janne Cadamuro

**Affiliations:** Department of Laboratory Medicine, Paracelsus Medical University, A-5020 Salzburg, Austria; j.cadamuro@salk.at; Tel.: +43-57255-57263

**Keywords:** machine learning, deep learning, neural network, tricorder, laboratory medicine, extra-analytics

## Abstract

Laboratory medicine has evolved from a mainly manual profession, providing few selected test results to a highly automated and standardized medical discipline, generating millions of test results per year. As the next inevitable evolutional step, artificial intelligence (AI) algorithms will need to assist us in structuring and making sense of the masses of diagnostic data collected today. Such systems will be able to connect clinical and diagnostic data and to provide valuable suggestions in diagnosis, prognosis or therapeutic options. They will merge the often so separated worlds of the laboratory and the clinics. When used correctly, it will be a tool, capable of freeing the physicians time so that he/she can refocus on the patient. In this narrative review I therefore aim to provide an overview of what AI is, what applications currently are available in healthcare and in laboratory medicine in particular. I will discuss the challenges and pitfalls of applying AI algorithms and I will elaborate on the question if healthcare workers will be replaced by such systems in the near future.

## 1. Introduction

Healthcare as a whole has evolved from a profession in which one had to acquire over years of training the ability to be able to interpret clinical signs correctly into a high-end field with tons of data to process. Clinicians are forced to review all this data, trying to formulate an initial orientation, sensing changes throughout the patient’s history and to make sense of these changes to be able to translate this convolution into medically actionable information. However, when considering that the input information may consist of anamnesis, clinical examination results, imaging diagnostics, laboratory reports, current medication, pathology reports, patient history including pre-existing conditions and even the individual’s social environment, it must become more than obvious that this amount of data is not processable by humans in a reasonable fashion. This illustration does not even acknowledge the fact that most of these pieces of information are often distributed across several sub-systems of electronic health record (EHR) systems or the fact that the formatting thereof often does not fit the purpose and vital information may be overlooked [1].

One would believe that more information is equal to higher quality. However, sadly the opposite seems to be true. By piling up more and more information the quality of medical care may decrease as clinicians are facing a so-called “information overload” [2]. Furthermore, the time spent on the task of gathering and making sense of patient data is dwarfing the time available for actual patient “care”. Brown et al., evaluated how physicians are reading medical patient progress notes and found that the average reading duration across all notes was 112 s [3]. Physicians spent a great deal of time reading the “Impression and Plan” zone of the notes (67% of total reading time), while “Laboratory Results” was screened for only 9 s. Even if laboratory reports are formatted in an ideal way, this obviously is far too little time for such an important part of patient evaluation. Therefore, we are in need of assistance, supporting medical professionals in data retrieval and processing, medical decision making and quality improvement by suggesting alternative diagnostic strategies. In my opinion and the opinion of many others, the only logical option must be the use of artificial intelligence (AI) [4].

## 2. But What Exactly Is AI?

The term “artificial intelligence” has come up as early as the 1950s. However, only in recent years has AI become increasingly of interest, as we now have the amount of data and the computational power needed to harvest the capabilities of these methods. In diagnostic medicine too, AI has been become increasingly of interest over the past years (Figure 1).

Therefore, let us first have a look at what AI stands for and what it is capable of. The main goal of AI is to perform complex operations in a human-like manner. This may reach from autonomously driving cars over smart home assistance to countless applications in health care. Eric Topol describes the era of AI, robotics and big data we are living in right now as the “fourth industrial age”, predicting that changes will be so profound that no previous life-changing invention would be eligible to compare it to, including, steam power, electricity or computers [6]. He believes that this revolution will take over all human endeavours, including medicine—but please do not be alarmed until you have finished reading the entire article. It is not as threatening as some Hollywood films want us to believe.

As AI only is a hypernym for several different models aiming at the same goal, let us look at those most used in medicine and many other fields so far, machine learning (ML) and a subset thereof, deep learning (DL).

ML describes an AI method in which a system is provided input data with or without the expected output and calculates or learns how to process this data to solve the problem it is faced with. This new methodology has taken the field of AI from exhaustive human programming efforts to self-learning machines.

ML can be subdivided into supervised learning algorithms like decision trees, unsupervised learning algorithms, such as clustering, and reinforcement learning algorithms under which Bayesian networks, neural networks and deep learning algorithms are listed. The most basic difference between supervised and unsupervised learning is that the former gets fed with label data, while the latter processes unlabelled data, trying to find hidden patterns. Reinforcement learning could be compared to a hit and trial method during which the systems discovers errors and rewards and learns what is true and wrong along the way, just like hitting the wall when trying to find the way through a labyrinth (Figure 2).

While speaking of models and algorithms, let us clarify their meaning in ML systems. Algorithms are a sort of program that can process different kind of data. A model is the product of task-specific model data that was processed by said algorithm. So, a decision-tree algorithm, fed with laboratory data of anaemic patients produces a model of specific if-then statements which may be used for root cause analysis of other anaemic patients. Hence, ML algorithms are never task-specific, while ML models are.

The structure we most often refer to when talking about laboratory diagnostic pathways would be very basic forms of decision tree algorithms, comprised of several “if > then” statements concatenated by and/or/not-logics. However, real ML decision trees are based on unselected input data including a label such as laboratory results, diagnosis and other patient data, calculating the possible paths to the correct Boolean, classification, association or numerical statement using deductive and inductive logic, while in “algorithms” programmed in to the laboratory information system (LIS), like ”*if TSH is high, measure fT4*”, the input data as well as the expected output are provided alongside a pre-defined path. Hence, such expert rule systems are not capable of self-learning or self-improvement and should not be mistaken for any kind of AI [7].

DL is an evolution of ML, expanding its capabilities. DL methods have shrunk the laborious feature engineering phase. Consequently sequential, spatial and temporal information existing in images, biosignals and anamnesis can be utilized in the training process more efficiently. It is basically taking a complex task and breaking it up into smaller and easier tasks (layers) then chaining those together to form a complete model. By breaking up the inputs into smaller parts, the DL algorithm is capable of recognizing patterns within these data. To accomplish this task, DL algorithms use so-called artificial neural networks which, as the name already suggests, aim to mimic the human brain and its neurons. Each “neuron” at the input level of such a network usually holds a number between 0 and 1 reflecting the “activation” of that neuron (e.g., the brightness of each pixel of an image). All of these numbers combined, based on the input data, make up the first “layer” of the artificial neural network (ANN). On the output level, a prediction model is presented in combination with the probability of the calculated prediction being true or not. All layers in-between the in- and output are called hidden layers whose function it is to “weigh” different characteristics of the input data and the combination thereof from layer to layer, identifying the most probable answer along the way (Figure 3).

For such task significantly more computational power is needed compared to most ML methods. The fact that this processing power now is available in combination with several open-source libraries and frameworks has made DL methods more popular in the past years.

The development of ML and DL algorithms can be divided into training and validation (see Figure 4). The majority of the ground truth data is used to train the algorithm. This is achieved by iterating and re-iterating and continuously adjusting/improving the algorithm, based on the comparison of the prediction model with the labelled data. Subsequently, the rest of the data is used to validate/test the algorithm to ascertain the model’s performance.

## 3. AI in Day-to-Day Life

In modern life AI surrounds us each day in various shapes and forms, sometimes or most often without us even knowing. With nearly every person carrying around a mobile tracking device, constantly connected to the internet, some extremely convenient features, facilitating all our lives, are being made possible.

In fact, everyone, including myself, is relying so much on their smartphones, that when the battery runs out or anything else happens that silences our phone, we are being left stranded. I myself was in such a situation when travelling to a congress to give a talk when my phone died near the airport. I did not know which flight I was booked on, which hotel to check in to and how to get there, I only knew country I was going to! This magnitude of ignorance is only possible due to the development and reliability of such AI systems and our way of life, taken these tools for granted.

We log into our smartphones or social media using our fingerprint or faces, we travel to countries we know nothing about, knowing we have a personal guide, a translator and every information on that country available with us all the time. Google even provides a tool which recognizes text in any image and translates it into any language in real-time. We can talk to any person using real-time speech translation tools. Online Clouds upload our images while we take them and categorize them by recognizing the content of the image in combination with the geotag of that image. YouTube, Spotify, Netflix, Amazon, etc. know exactly who we are by analysing our habits using AI algorithms, making personalized recommendations fitting exactly to our preferences. When typing a search term into Google, an AI algorithm predicts what you are searching for and provides suggestions for auto-completion.

In recent years, so-called smart home devices and digital assistants, which five to ten years ago were tools only used by technophile nerds, blended into our way of life as a natural and convenient supplement. These systems heavily rely on AI algorithms, checking the user’s position, depending on day time, temperature, habits, preferences any so many more variables, providing the optimal response or outcome to each situation.

Apart from these systems aimed at making our lives easier, there are lots of AI solutions for nearly every situation one could think of, like generating licence-free music (https://aiva.ai (accessed on 10 May 2021)), generate photos of non-existing people for your next presentation (https://generated.photos (accessed on 10 May 2021)), writing essays, (https://www.essaybot.com (accessed on 10 May 2021), https://www.shortlyai.com (accessed on 10 May 2021)) or more creative tools for creating an old painting of your images (https://ai-art.tokyo (accessed on 10 May 2021)) or to animate images of your grandparents (https://www.myheritage.at/deep-nostalgia (accessed on 10 May 2021)).

Last year, “Generative Pre-trained Transformer 3” (GPT-3), the largest artificial neural network ever created was released [8]. GPT-3 is a powerful language prediction model, hence it can create anything that has a language structure, like writing emails, answering questions (e.g., as chat bot on websites), write essays, summarize long texts, translate languages, take memos, it can write news reports, creative fiction, it can even create computer code for an entire app based on the users input in plain text on how it should look. As one of the major drawbacks was that this network is available only in English, other similar networks have been released since, like the Chinese PanGu-Alpha library [9].

Similar to the 1990s when no one would have predicted our way of living today, it is near to impossible to predict how our life will change over the next twenty years. One thing is for sure, however: AI will play a huge part.

## 4. AI in Medicine Today

The difference between AI solutions presented in the previous chapter and AI in health care are the consequences of error. In patient care an error made by one physician may cause harm to one patient, while an error made by an algorithm aiding in medical decision making may harm far more patients. Furthermore, an accuracy of 90% may be sufficient for convenience applications, meaning that, i.e., one out of ten voice commands is not being understood correctly by smart home devices. In medical care, however, it translates to a 10% inaccuracy, which would be regarded as a major threat to patient safety. Therefore, technology must be under strict regulatory oversight and comply to legal requirements such as the FDA 21 CFR Part 11 to prevent loss of accuracy [10]. What this directive basically requires is the authentication of the validity of electronic records, confirmation of the authenticity of electronic signatures and records and the validation of the reliability of electronic records and signatures. Translated to healthcare this implies using validated equipment and computer systems, secure retention of electronic health records to instantly reconstruct analyses, user-independent, computer-generated, time-stamped audit trails, system and data security, data integrity and confidentiality through limited authorized system access as well as secure electronic signatures for closed (under the control persons with restricted access) and open (all other including data transfer over the internet) systems [11]. Or in simpler words: it does not allow “black-box” systems such as deep learning algorithms. This, of course is a major obstacle in the endeavour to better diagnostic assistance. It applies whenever (medical) information in text, images, video or audio is to be electronically generated, amended, stored, transferred or accessed.

Additionally, if that would not be enough there are also other regulations medical algorithms have to comply with like general data protection regulations (GDPR) [12], the medical devices regulation (MDR) [13] or the in vitro-diagnostica regulation (IVDR) [14] and others. These and other regulations may differ between countries, but the overall complexity in aiming to implement machine learning or even deep learning algorithms into laboratory diagnostic practice stays the same.

All of this said, we should acknowledge the fact that there is another reason for medicine lagging behind the rapidly evolving technological progresses in our daily lives. Medicine is traditionally quite reluctant to change and to adoption of new technologies, or as Eric Topol puts it: “*Health care is still stuck in the first part of the third revolution, the digital transformation*” [6]. Additionally, even if electronic systems are in use, these are often not compatible among each other due to proprietary issues, leaving the clinician to scrape together all parts of patient data from different subsystems, trying to make sense thereof and to see the bigger picture or quoting once more from one of Eric Topol’s books: “*Your ATM card works in outer Mongolia, but your electronic health record can’t be used in a different hospital across the street*” [6]. However, first attempts have been made to apply deep learning techniques in order to sift through patient documents within an EHR, collect clinically relevant data aiming to find common diseases [15].

Two decades ago, nobody would have foreseen that photo camera, video camera, calculator, calendar, phonebook, internet, recorder, telephone and much more would fit in one small device and that the majority of people on earth would own one. As my mathematics teacher used to say, *“You need to know mathematics, since you won’t be running around with a calculator in your pocket all the time”*. Well, in your face Mr. Gardner! Similarly, the ultimate goal of diagnostic healthcare should be inspired by the medical tricorder from the Sci-Fi series *Star Trek* (Figure 5).

A small device which can detect all deviances from the norm in a human body as well as its root cause without even touching the patient. All of the efforts we put into AI algorithms, assisting physicians in diagnosing the patient are just steps in that direction. We may consider this vision as a motivator for a continuous improvement process. Indeed, there even was a competition named Qualcomm Tricorder XPRIZE, which would pay USD 10 million “*to incentivize the development of innovative technologies capable of accurately diagnosing a set of 13 medical conditions independent of a healthcare professional or facility, ability to continuously measure 5 vital signs, and have a positive consumer experience*” [16]. USD 4.7 million of this prize was ultimately awarded to promising projects.

Currently, most FDA approved AI algorithms implemented in healthcare are based on pattern recognition in images or electrocardiograms [17]. Therefore, it is hardly surprising that medical specialities involving image interpretation in the diagnostic process are the major target of such systems. Aggarwal et al., who recently published a systematic review and meta-analysis on the topic of deep learning in medical imaging, found an AUC ranging between 0.933 to 1, 0.864 to 0.937 and 0.868 to 0.909 for selected diagnoses in ophthalmologic, respiratory and breast imaging, respectively [18]. Annarumma et al., tested deep convolutional neural networks (CNN) to triage chest radiographs from adults based on the urgency. Normal chest radiographs were detected with a negative predictive value (NPV) of 94% and a positive predictive value (PPV) of 73% (sensitivity of 71%, specificity of 95%) [19]. Several companies already provide such systems to aid radiologists in interpreting X-ray-, CT-, MR and other images, one of which even claiming to currently serve 30% of US hospitals with AI powered tele-radiology solutions [20].

Another medical speciality in which AI solutions are rapidly evolving is the field of dermatology. Four years ago, Esteva et al., constructed a single CNN for the detection of skin cancer and found an AUC of 0.96 for carcinoma and of 0.94 for melanoma detection, on par with expert opinions [21]. Even smartphone apps are available with a sensitivity of 80% and a specificity of 78% for the detection of malignant or premalignant lesions; however, that have poor accuracy compared to expert recommendations [22].

Additionally, in ophthalmology, as already mentioned above, several AI techniques have been studied with an astounding level of accuracy [23,24,25]. Equally impressive are data from a study assessing the potential of AI to detect adenomatous or sessile polyps during live colonoscopy [26]. Furthermore, many of the currently FDA approved ML algorithms are designed to aid in cardiology diagnostics, either by detecting and monitoring patients for arrhythmias, interpreting electrocardiograms or cardiac imaging interpretation [17,27]. In pathology, pattern recognition algorithms on histologic slides have proven to be useful for the detection of cancer cells [28,29].

In paediatric care, where information about the child’s symptoms usually is obtained second-hand from their parents, diagnostic tools using AI could be of tremendous help, not only for the diagnostics itself, but for telemedicine, making devices so smart that parents can use it on their child and send the data to the physician [30]. Several technologies are currently being investigated such as a smartphone-based otoscope attachment with the ability of assessing eardrum mobility by the use of the phone’s speaker and microphone, detecting middle ear fluids [31]. Other strategies include an AI classification of a child’s coughing sound [32], monitoring a baby’s jaundice severity using the images of its skin [33] and photos of fingernails as a needle-free alternative to detect anaemia in children [34].

According to the Stanford Medicine’s 2020 Health Trends Report, healthcare providers are now adapting to these new developments, with nearly half of all physicians and three quarters of medical students currently seeking out additional training to better prepare themselves for innovations in health care with 34% of which pursuing classes in artificial intelligence [35]. However, there still seems to be a transformation gap among current and future physicians regarding their readiness to implement emerging technologies. Since medical curricula are very reluctant to change and have not yet adopted to new and emerging situations, not very surprising, only 18% of current medical students and residents surveyed in this report said that their education was “very helpful” and current medical professionals still feel insufficiently trained to bring new technologies and insights to the patient bedside.

The outcome of AI algorithms is only as good as the input. Hence, lots of high-quality data is needed, which is why several electronic device manufacturers have started developing wearables, able to collect such data in real-time. Some of which have already been approved by the FDA as a medical device, such as the Apple Watch Series 4 for the detection of atrial fibrillation [36]. Other data that can be measured and documented for further use by some of these devices are vital signs like blood pressure, heart rate and rhythm, blood oxygen saturation, respiratory rate and temperature and of course any kind of activity protocol using a built-in gyroscope and GPS technology. This new technology holds invaluable possibilities not only for the detection of diseases at an early stage but also for remote diagnostics and monitoring.

## 5. AI in the Medical Laboratory

The main task of each medical laboratory is to measure specific analytes within the patient´s specimen, so that these data in combination with other diagnostic and clinical findings can be translated into clinically actionable information. This demand was met by laboratory specialists in the past, however, as laboratory medicine evolved rapidly over the past decades, the data needed to be processed increased exponentially, forcing laboratory specialists to filter their data and focus on selected patients only (i.e., patients with results severely outside the reference range). Hence, clinicians are being left alone with test selection and interpretation, which in turn led to an over- and underuse of laboratory resources, including a potential patient risk by deviating from the principle of the five rights (the right test in the right patient at the right time using the right sample and right transportation) [37,38,39,40,41].

As Mrazek et al., elaborate in this issue, there are several ways to overcome this issue, with laboratory diagnostic algorithms being one of them (CROSSREF). Until now these algorithms were expert rule systems implemented into the LIS, a concept known since 1984 and often referred to as Computerized Clinical Decision Support Systems (CCDSS) [7]. However, with the availability of huge amount of well-structured patient data in combination with the increased computational power, laboratory medicine is a perfect playing ground for the development of AI models. However, currently only the minority of FDA approved systems include laboratory diagnostics [17,42].

AI algorithms could aid in many steps of the total laboratory process. In the following sections I will provide some selected examples.

### 5.1. Test Selection

Failing to identify the appropriate test for the individual patient’s symptoms is one of the major contributors to laboratory over- and underuse. To date, not many studies have been published on AI models recommending laboratory tests for individual patient care. The first study to do so was published in 2020 by Islam et al. [43]. The authors developed a DL algorithm as recommendation system for laboratory tests. They performed a retrospective analysis of 129,938 lab orders of cardiology patients, retrieved from their national health insurance database. Considering 1132 input variables, the model was able to predict 35 types of laboratory tests with an area under the receiver operating characteristic curve (AUROC) range of 0.63 to 0.90, with the need of only minimal patient input data such as gender, age, disease and drug information.

In this issue the same authors present a similar DL model for personalized laboratory test prediction, which was developed using 1,463,837 lab orders from 530,050 unique patients [44]. This model achieved an AUROC of 0.92–0.96 for 114 laboratory tests, 0.96–1 for 106 laboratory tests, and 0.88–0.92, 0.84–0.88, 0.80–0.84 and 0.76–0.80 for 56, 30, 5, and 4 laboratory tests, respectively.

Cheng et al., applied a similar approach by retrospectively evaluating lab tests from an ICU, identifying those orders which actually led to a change in patient care [45]. The authors developed a framework for clinical decision support, focussing on laboratory tests used for the diagnosis and follow-up of sepsis and renal failure. The two-fold approach tackled by a reinforcement learning (RL) based method consisted of building an interpretable model to forecast future patient states and modelling patient trajectories as a Markov decision process. The authors found an estimated reduction of 27–44% of lab orders, a higher mean information gain and calculated that recommended orders happened one to four hours earlier than the actual time of an order by the clinician.

Similarly, Xu et al., aimed to identify potentially superfluous lab tests by applying ML models, including regularized logistic regression, regress and round, naive Bayes, neural network multilayer perceptrons, decision tree and random forest algorithms [46]. The models should predict normal results of lab orders as defined by local laboratory reference ranges in the retrospective data of 116,637 inpatients. The best-performing model did so with an AUROC of 0.90 and above for 22 laboratory tests.

### 5.2. Predicting Test Results

Despite predicting which lab tests will hold no clinical value in the individual patients, other models have been developed to estimate the test result. Luo et al., used several machine learning algorithms to predict ferritin levels, based on the patients’ demographics and other test results from haematology or clinical chemistry, reaching an AUROC of 0.97 [47].

Lobo et al., sought a solution to predict one-, two- and three-month haemoglobin (Hb) levels in patients with end stage renal disease after being treated with erythropoiesis stimulating agents in order to improve their therapeutic strategy [48]. They developed a recurrent neural network approach, based on historic patient data in combination with future therapies predicting the patients Hb values with a mean error rate of 10–17%. Another approach of predicting Hb results was published by Mannino et al., who developed a smartphone-app, able to estimate Hb levels on the basis of pictures taken of the patients’ fingernails [34]. The authors show that their system can detect anaemia with an accuracy of ±2.4 g/dL and a sensitivity of 97%.

Burton et al., used an ML approach testing three algorithms predicting negative culture results in patients with suspected urinary tract infection in retrospective data [49]. The authors used independent variables, including demographics, historical urine culture results and clinical details provided with the specimen. The best performing algorithm would have been able to reduce the workload by 41% with a sensitivity of 95%, which is why the authors decided to implement this model into clinical practice. Yasin et al., developed calculations, able to derive potassium levels from the patient’s electrocardiogram (ECG) with an error rate of only 9% [50]. Yu et al., aimed to reduce unnecessary repeated lab tests, based on observed previous testing values, while maintaining the maximum precision in patient diagnostic testing [51]. They developed a deep neural network (DNN) model, able to predict which repeated tests could be omitted and estimate its result, leading to a calculated 15% reduction in blood draws with an accuracy of 95%. Similarly, Dillon et al., published as early as 2015 a model that is able to determine potassium levels from electrocardiograms (ECGs) with correlation coefficients between 0.77 and 0.99, detecting changes in potassium levels by as little as 0.2 mmol/L [52].

Of course, all these algorithms can provide a good estimation at best, but this might be sufficient in some clinical settings.

### 5.3. Result Generation

Opposed to predicting test results based on individual patient variables, some other systems aim to actually generate laboratory test results. The most obvious method hereof is applying image recognition software to identify corpuscular specimen components. Such systems have been in use for some time now for the identification and quantification of blood cells or urine sediment components [53,54]. However, one major drawback of these instruments is their inability to constantly learn and improve or to be taught by the user. A new system combining imaging and ML algorithms claim to be able to detect and enumerate malaria species within erythrocytes [55]. However, to date no published evidence is to be found critically evaluating this system.

As early as in 2008, Atliner et al., evaluated a software, consisting of six associative neural networks, which would differentiate between normal and anomalous capillary electrophoresis patterns, finding that this system is comparable to human experts [56].

Other approaches are based on detecting colour-gradients, like the first FDA approved smartphone-app for urine dipstick analysis [57] or the so-called “Bili-Cam” app for monitoring newborn’s jaundice [33]. The latter uses the smartphones camera and a reference colour scale, yielding a 0.85 rank order correlation with the gold standard bilirubin blood test.

### 5.4. Result Interpretation

One task where AI solutions would be extremely helpful is gathering all existing patient information and transforming it into clinically actionable interpretation. Laboratories are usually struggling with this task as clinical patient information is mostly not provided alongside the lab order. There are, however, some areas where interpretation of test results is possible also without those data, i.e., HbA1c, cardiac troponins, creatinine and eGFR, infectious or haematological diseases and other. Gunčar et al., used this fact to develop a random forest (RF) model, predicting haematological diseases trained with 61–181 routine blood tests and depicting the ten most probable diseases as polar chart [58]. Considering the first five predictions, the models accuracy reaches 86–88%. The authors compared their model to haematology specialists, where the former achieved an AUC of 0.60 and 0.90 and the latter of 0.62 and 0.77, when taking only the first or the first five diagnoses into account, respectively.

Wickramaratne et al., developed a DL model using Bi-Directional Gated Recurrent Units processing and interpreting several laboratory test results from biochemistry, haematology, blood gas analytics and coagulation as well as vital signs [59]. This model was able to predict sepsis onset up to 6 h ahead of time with an AUROC of impressive 0.97.

A new emerging field in laboratory diagnostics is the field of genome, transcriptome, proteome and metabolome. Evaluating data from these –omics will help identifying biomarkers of disease by gaining insight into mechanisms underlying the complex development of diseases and by sub-phenotyping oncology patients or unscrambling the metabolic syndrome. However, resulting data are extensive and complex and impossible to interpret without computational aid [60,61].

### 5.5. Other Applications

Apart from all of the mentioned applications, additional strategies for implementations in medical laboratories include prognosis prediction or preanalytical improvements. One very interesting approach by Mayampurath et al., was to transform clinical patient variables into a graphical output along a time axis in order to predict in-hospital mortality [62]. The authors developed a CNN with a recurrent layer model, using data such as vital signs, laboratory results, medications, interventions and others from 115,825 hospital admissions, resulting in a prediction model with an AUROC of 0.91. Other examples for predictive modelling in the clinical laboratory have recently been put together by Naugler et al. [63].

Another interesting application of machine learning algorithms in medical laboratories was published by Mitani et al., who aimed to overcome the routinely used delta-check method to detect preanalytical specimen mix-up [64]. Using fifteen common analytes from haematology and biochemistry, the authors found an impressive AUROC in detecting preanalytically compromised specimen of 0.9983.

## 6. Challenges and Pitfalls

Nearly all of the studies currently available on the topic of AI in laboratory medicine are of retrospective nature using ROC, positive and negative predictive values (PPV, NPV) as metrics for operational value of algorithms. Only few have been reported to have been implemented in routine diagnostics [49]. Until prospective studies are available, findings of published algorithms need to be interpreted with respectful caution.

Apart from these, there are several other variables needed to be considered when evaluating the quality of proposed models. The most important would be the data used for training and testing. Ideally, this data is machine-readable, present in a “atomized”, structured and standardized way with no or only minor gaps within the dataset. In order to use datasets from different laboratories, analytes need to have an unambiguous designation, adhering to international terminologies. More on this topic can be read in the article of Bietenbeck et al., within this issue (CROSSREF). The more data that are available to train a model, the better the quality thereof. This might be the most limiting factor in modelling algorithms in healthcare, since much of these data need to be provided by humans and may therefore be biased or only inconsistently available. An example of the consequences of using only a small number of synthetic, non-real cases with very limited input is the steep fall of IBMs prestige project IBM Watson [65]. This system claimed to ingest millions of pages of medical information and transform these into treatment recommendations for cancer patients. This USD 62 million project ultimately collapsed, failing to process the unstructured data, acronyms, shorthand phrases, different writing styles and human errors within the datasets [6].

Whenever aiming to evaluate the quality of any AI model both training and test performance thereof should be recognized. Low training and high-test accuracy may indicate that the same test set has been used too many times. In case of small data samples, in particular, it should also be confirmed that the findings are different to chance level by, e.g., permutation tests.

However, in modern medical laboratories, orders are received from the EHR and processed by the LIS in a defined and structured way, which is why all of the requirements for data quality are met. When able to connect this information with other clinical patient data, sorted by current and pre-existing conditions, the opportunities for AI supported process improvements are limitless. However, before falling into euphoria, it has to be acknowledged that country-specific data protection regulation must be adhered to, an obstacle to which many good ideas already have been fallen victim to. Additionally, the very important issue of human made endpoints (i.e., diagnosis), which the model is fed to learn on has to be considered. These diagnoses do not necessarily be the correct ones or based on other data, not contained in the dataset. Obviously, the resulting algorithm might be of minor quality, again stressing the concept of “garbage in–garbage out”. Good models need clean and consistent ground truth data. These pitfalls need to be closely monitored in ML models, including the ones mentioned in Section 5.

After its acquisition, the data needs to be filtered for the intended use. However, care has to be taken as this filtering may obscure additional unexpected findings or may result in lower model quality as the example of the aforementioned study of Dillon et al., shows [52]. The authors teamed up with a company called AliveCor to develop a smartwatch-based system to determine potassium levels from the recorded ECG. However, when they used the ECG- and potassium data, provided by the Mayo Clinic, they only achieved an AUC of only 0.63. After months of investigation and remodelling, they had the idea of using data from all patients instead of the data that has been filtered on out-patients only. Doing so, they had a much larger scale of potassium values and additionally, ECG and potassium measurements were timely closer together in in-patients. Additionally, they broadened their model to focus not on the T-wave only, but on the entire ECG. These interventions resulted in an increase of the AUC to 0.86 and the authors and AliveCor moved on seeking for FDA approval [6]. This example holds several important lessons. Firstly, if you filter your data too early, you might end up decreasing the quality of your model. Secondly, do not assume that human-made facts (i.e., that only the T-wave holds the important information on potassium levels) are unalterable. If you feed a model with enough data in the rawest kind, it should be able to apply according filters by itself.

Another important issue is the way how AI models are dealing with outliers. When the dataset is “smoothened” by excluding respective entries, the resulting model may not reflect the entire population, potentially identifying patient slightly differing from the norm as abnormal, similar to laboratory results slightly above or below the reference range. The only way to meet this potential drawback are physicians, recognizing these models as the assisting tools they actually are, programmed to aid in medical decision-making for the majority of patients, but surely not all of them. A good example of how to apply these thoughts on non-medical AI models is the afore-mentioned website where you may generate photos of non-existing people. While browsing all the rendered faces, you will not an ugly, unique or explicitly memorable one that could be defined as an “outlier”, but which we see every day in real life.

## 7. AI Applied Incorrectly

Without understanding which data is being processed to form an automated computer assisted hypothesis, results could be misinterpreted or misclassified [66]. In daily life, this may result in incorrect suggestions for films on Netflix or goods on Amazon at best and in worst case scenarios it could lead to misuse by generating so-called deep-fake videos or purposely manipulating masses of users like in the case of Cambridge Analytica [67]. A frightening simulation on the future of AI in the weapon industry has been released by the Future of Life Institute in order to raise awareness [68].

When applied on medical diagnosis or treatment decisions, consequences of erroneous algorithms may always jeopardize patient safety. So-called online symptom checkers have been emerging during the last decade, where patients seeking advice about an urgent health problem can input their symptoms via a website and receive a diagnosis, a treatment and/or triage advice and in some cases even a medication prescription. Some of these symptom checkers claim to have doctors at the backend, others rely solely on AI algorithms. Although these internet-sites state that their tools are not a substitute for professional medical advice, diagnosis or treatment, and that after you found your most probable diagnosis you should check it with your doctor, it is doubtful that patients really comply with these advices. In 2015, Semigran et al., evaluated 23 online symptom checkers, using 45 standardized patient vignettes and found that these systems provided the correct diagnosis as first suggestion in 34% of cases and listed the correct diagnosis within the top 20 diagnoses in 58% of cases [69]. Triage advice was appropriate in 57% of these evaluations. Four years later Chambers et al., performed a systematic review following the same goal [70]. They too found that the data collected from 29 publications demonstrated a generally low diagnostic accuracy. The authors also found that these online services are being used especially by younger and more educated people.

Apart from symptom checkers, whenever using AI algorithms for medical decision-making or support thereof, it should be acknowledged that these programs can replicate institutional and historical biases, amplifying disadvantages lurking in data points, such as health status ratings or scores [71]. As initially mentioned, the dataset is the key for high-quality algorithms. Since these data sets may be riddled with gaps and some models developing algorithms in hidden layers without the possibility of reconstructing the modelling process, care needs to be taken whenever relying on algorithmic decision-making.

## 8. Will AI Replace Us?—No!

In a recent Reddit discussion, former chess world champion Gary Kasparov, who famously introduced the world the power of AI by losing a game to Deep Blue, was asked what he would think the most common misunderstandings of AI was today [72]. He answered: ”*To pick the biggest one, it’s that AI is a threat instead of a powerful technology like any other that is agnostic, and good or bad depending how we use it. It’s a very harmful outlook, because we need to be more ambitious and more optimistic so we invest more, learn more, and get the benefits, not just suffer the slow-moving consequences of disruption and automation”.*

We tend to believe that AI may at some point develop self-awareness, imagination, manipulation, feelings and empathy and therefore may pose as major threat to human kind as many Hollywood films like *Space Odyssey, Ex Machina, A.I.* or *I, Robot* want to make us believe. However, AI algorithms are only as good as the data input. There will be no replacement for a physician’s intuition after many years of experience. AI is or will be able to perform monotone routine tasks with defined in- and output and provide rapid information on static knowledge. As this may be the definition of many lower skill working environments, such jobs may be at higher risk for automation [73]. On the other side, tasks that are not very likely to be replaced by AI are non-routine tasks, requiring critical thinking and creative problem-solving, which basically describes the medical profession or at least parts thereof. Therefore, we should embrace new technologies assisting us with the (non-medical) routine tasks. Sinsky et al., found that physicians in an ambulatory setting are spending 27.0% of their time on direct clinical face time with patients and 49.2% of their time on EHR and desk work [74]. Implementing AI technology as a symbiotic assistant could free more of the physician’s time, which could then be used to refocus on the main profession, start listening to the patient again and examining him or her in detail instead of looking only at screens and treating numbers.

Additionally, computer-based algorithms are not able to replace the physicians visual, auditory, tactile, olfactory perception. The idea of the physician’s intuition is based on the Gestalt-principles, which basically states that even if such IT systems perform better than humans in defined tasks, the integration of individual clinical expertise and judgment of an experienced physicians will always improve that outcome [75]. This theory has been proven in the past, i.e., by Wang et al., who found that the ROC of an AI algorithm detecting breast cancer cells was 0.925, while for pathologist it was 0.966, but in combination the ROC increased to a staggering 0.995 [29].

In terms of laboratory data acquisition, AI systems, implemented on handheld devices such as smartphones, will most probably evolve within the next year, becoming a kind of point-of-care-testing 2.0.

## 9. Conclusions

The coming years and decades will be dominated by AI-assisted improvements in our day-to-day life as well as in healthcare. The possibility to collect all kinds of patient data in real time using smartphones or wearables as well as the ever-increasing analytical possibilities of medical laboratories are producing data in an extent which is not processable by humans alone. Very promising models have been developed in recent years, but only very few have reached clinical practice. This will most probably change rapidly and will forever change the way physicians work, freeing up valuable time for actual patient care. It is an inevitable evolution and we should embrace rather than condemn it and learn how to use it most efficiently.

However, before applying such models in clinical care, a thorough evaluation of their training and test performance has to be performed in order to assure reliability and safety for our patients. Additionally, these tools have to be acknowledged as assistance at best. They should not be used carelessly and always in conjunction with the physicians or laboratory specialist’s clinical expertise.

Additionally, all these thoughts and brilliant ideas are hollow talk, when considering that large hospitals in central Europe do not yet provide wireless networking and subsequently no electronic mobile patient health record, the standard web-browser is the Internet Explorer and laboratory tests are mostly ordered using non-evidence-based test-panels.

There is a long way ahead of us but every journey, no matter how far, begins with the first step. Medical professionals will need to adapt to AI technologies as their new partners in patientcare. This is especially true for laboratory diagnostics, as it is a field comprised of structured numbers and their interpretation which is already today contributing to the majority of medical decision-making.

## Figures and Tables

**Figure 1 diagnostics-11-01399-f001:**
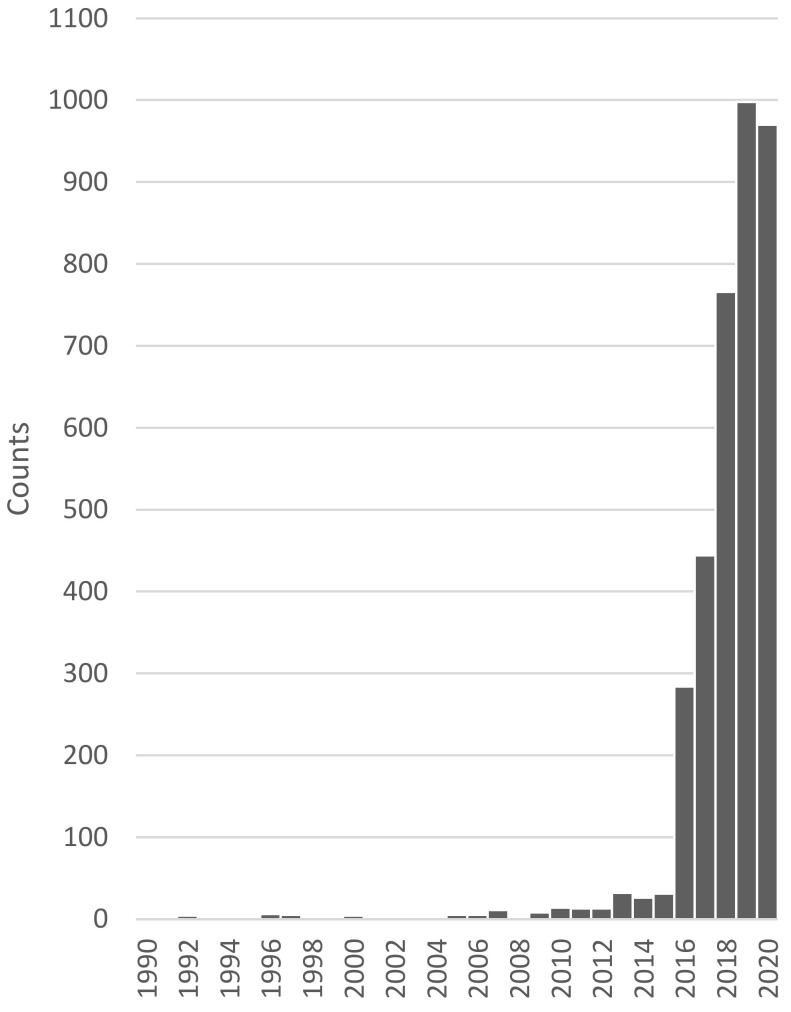
Number of news articles, publications and conferences connected to the topic of AI in diagnostic medicine over time. Data was collected and is provided by the Association for the Advancement of Artificial Intelligence [5].

**Figure 2 diagnostics-11-01399-f002:**
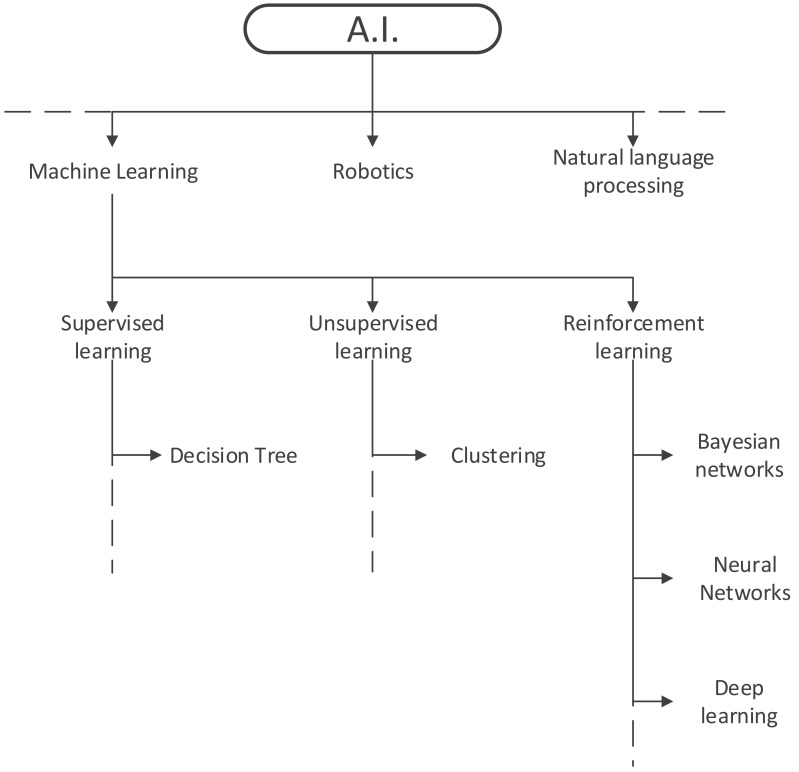
Subtypes of artificial intelligence.

**Figure 3 diagnostics-11-01399-f003:**
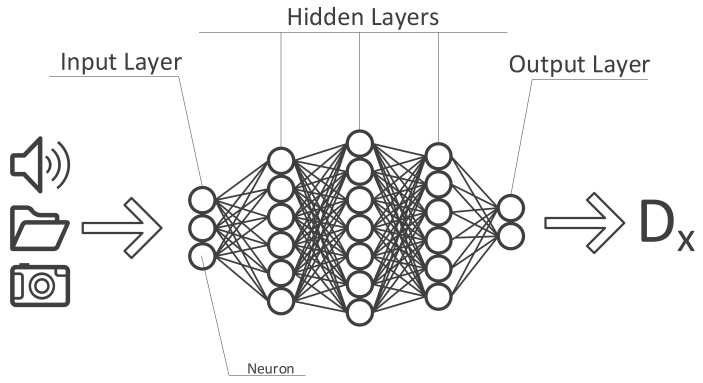
Structure of an artificial neural network.

**Figure 4 diagnostics-11-01399-f004:**
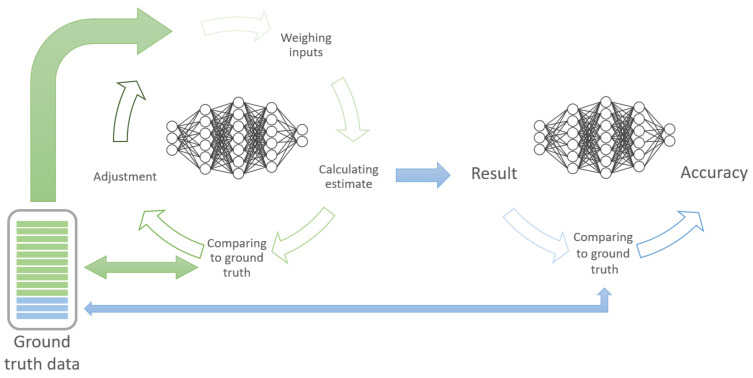
Concept of training (green) and testing/validating (blue) ML models.

**Figure 5 diagnostics-11-01399-f005:**
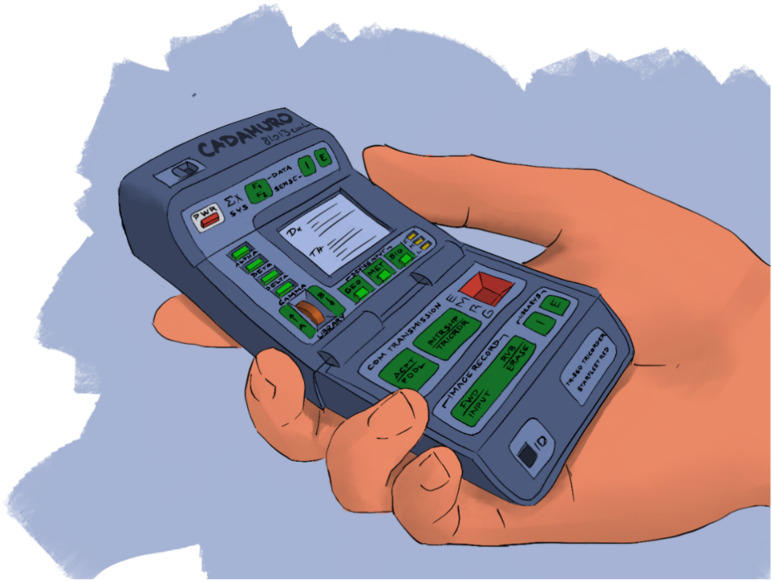
The medical tricorder from *Star Trek*—The future of medical diagnostics?

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
