# Peer review of "Rise of the Machines: The Inevitable Evolution of Medicine and Medical Laboratories Intertwining with Artificial Intelligence—A Narrative Review"

_diagnostics, 2021, doi:10.3390/diagnostics11081399_

Round 1

Reviewer 1 Report

I would like to congratulate the author on a well thought-out and written exposition of the history, current-state, and potential future for the use of AI/ML in health-care and specifically in laboratory medicine (pathology laboratory).

It was a relief to see that that the author has not been excessively seduced by the buzz-words of Artificial Intelligence, Machine Learning etc. In particular to recognise and remember that they feature very little of what we consider as human Intelligence and human Learning. While there is a risk that being sceptical of these technologies will slow-down their adoption in health-care settings, the alternative - for infant technologies to dazzle and enchant health-care administrators into premature adoption and resulting in an Emperor's-new-clothes scenario of self-deception - is much worse. In this light I was glad to see examples provided of AI/ML models not working well, or in the case of IBM Watson, actually failing.

If there was one issue/topic that I was hoping to see mentioned/addressed, it was the issue of dealing with outliers, both in how AI/ML models are trained but also outliers in the population where you hope to actually use the AI/ML based tool. Just because you are slightly different to the normal population, you wouldn't want to be recommended a bad treatment! Probably the response to this is that this is the reason we still include the physician in the health-care process - they are able to actually think! In any case, it would be nice to see the issue explored a little bit. 

To extrapolate the outlier scenario above and link it back to things discussed in the manuscript: generated.photos website contains millions of synthetic faces, but despite scrolling for several minutes, I struggled to find any faces that I would consider to be ugly or very unique (and unfortunately, there is no search function to search for ugly) but when I catch public transport to and from work, I see people along the entire spectrum of very ugly to very attractive (or similarly, faces that are unique and memorable).

I notice one or two places that the word CROSSREF appears in the m.s., I assume this is just some referencing software having played up, and will be fixed before the final version.

Author Response

Comments and Suggestions for Authors

I would like to congratulate the author on a well thought-out and written exposition of the history, current-state, and potential future for the use of AI/ML in health-care and specifically in laboratory medicine (pathology laboratory).

It was a relief to see that that the author has not been excessively seduced by the buzz-words of Artificial Intelligence, Machine Learning etc. In particular to recognise and remember that they feature very little of what we consider as human Intelligence and human Learning. While there is a risk that being sceptical of these technologies will slow-down their adoption in health-care settings, the alternative - for infant technologies to dazzle and enchant health-care administrators into premature adoption and resulting in an Emperor's-new-clothes scenario of self-deception - is much worse. In this light I was glad to see examples provided of AI/ML models not working well, or in the case of IBM Watson, actually failing.

I am grateful that the reviewer is sharing my thoughts on this topic and I thank him/her for these motivational words.

If there was one issue/topic that I was hoping to see mentioned/addressed, it was the issue of dealing with outliers, both in how AI/ML models are trained but also outliers in the population where you hope to actually use the AI/ML based tool. Just because you are slightly different to the normal population, you wouldn't want to be recommended a bad treatment! Probably the response to this is that this is the reason we still include the physician in the health-care process - they are able to actually think! In any case, it would be nice to see the issue explored a little bit. 

To extrapolate the outlier scenario above and link it back to things discussed in the manuscript: generated.photos website contains millions of synthetic faces, but despite scrolling for several minutes, I struggled to find any faces that I would consider to be ugly or very unique (and unfortunately, there is no search function to search for ugly) but when I catch public transport to and from work, I see people along the entire spectrum of very ugly to very attractive (or similarly, faces that are unique and memorable).

This issue indeed is an important one and I thank the reviewer for bringing this to my attention. I added a respective paragraph in the “Challenges and Pitfalls” section.

I notice one or two places that the word CROSSREF appears in the m.s., I assume this is just some referencing software having played up, and will be fixed before the final version.

Yes, these crossreferences are placeholders for references on articles in the same special issue.

Reviewer 2 Report

The aim of the manuscript is provide an narrative review of what AI is, what application are available in healthcare and laboratory medicine. Moreover, the author's is to discuss challenges and pitfalls of AI, especially about the concern of whether the humans will be replaced by AI systems in near future.

The topic of the manuscript is important and timely due to prevailing AI hype during the recent years.  The paper also refers to some unrealized AI promises by industry that have led to both unrealistic expectations and fears in the fields of health care and medicine.

The paper is also well-organized and easy to read. Scientifically the manuscript is on a rather general level. It is clear and comprehensive, but does not provide much scientific facts concerning performance, weaknesses, challenges of AI methods for researchers. Therefore, the manuscript might be more relevant in some popular science journal, but as a narrative review can be considered relevant in the applied journal.

I present some comments that I hope the author consider before the final acceptance. My review is primarily based on the machine learning expertise with a lot of collaboration with medical and health care specialists. Therefore, my comments are mainly related to machine learning.

Chapter 2.

  • I would elaborate the description about AI and ML. For this paper it is important describe more precisely the difference between classical AI versus ML based AI that has taken the field from exhaustive human programming efforts to learning machines. The algorithms are not anymore task-specific, but the models are since the learn from the given data. It might be also good to explain most essential terms such as learning algorithms (the general algorithm that fits the models to the data), model and training, validation and testing data.
  • Furthermore, it is important to describe difference between ML and DL. The major difference between the two is the laborious feature engineering phase whose role has shrunk due to deep learning methods. Consequently sequential, spatial and temporal information existing in images, biosignals, anamnesis can be utilized in the training process more efficiently. 
  • Deep learning has become popular due to the available open-source libraries, GPUs processors and adequate raw data (e.g. images)
  • Deep learning and related technological advances make things easier for data scientists, but what I always tell to medical specialists is that the modern AI models are still highly task specific. They can only what they have seen in the data. So there is really no reason to worry about machines roles.
  • Reconsider the sentence "Each "neuron" at the input level of such a network holds a number between 0 and 1 ...." The number depends on the type of preprocessing, it can also be less than 0 or greater 1.
  • The sentence "On the output level..." is unclear. Moreover, it does not take into account regression models in which the output(s) can be any real numbers.
  •  Table 1 is questionable
    • I do not agree that performance of ML model does not increase with the amount of data. I think it increases for both, but due to the greater learning capacity the performance of DL models saturate later.
    • I don't understand the sentence: "Can function with a small amount of data"? 
    • ML methods may also require a lot of computational power, for example, training a large number of classical MLPs on small sample EEG data may require randomization for avoiding chance models. This may require a huge amount of computation.
    • "Tracing back the problem solving path" This is true for some traditional methods, but not necessarily for all, for instance classical MLPs, SVMs,,
    • The four statements in Table 1 should be also supported by some references.

Chapter 3

  • I think the content of this chapter is mainly irrelevant for the manuscript

Chapter 4

  • Line 276: The sentence "How efficient this device is ...." is inappropriate for a scientific paper. It sounds like marketing without any real evidence. A random observation is not evidence of anything.

Chapter 5

  • Without any information about training, validation and testing process/strategy and related data sampling the performance measures are difficult to interpret, but can be sufficient for a narrative review 

Chapter 6

  • Data quality (noise, erroneous and incomplete data) is always an issue. Since filtering and pruning data is also a risk (of distorting data distribution),  I would suggest to discuss also about robust learning methods that tolerate noisy and even erroneous data without filtering by utilize robust statistical estimates. Moreover, most of the data are incomplete (part values are missing) for which methods based on preprocessing free strategies such available case strategy could be considered. 
  • Inconsistent ground truth data is also an issue. For instance, a large number of DL models have been developed to perform K-L grading for knee X-rays, but without real success in early detection. It is thereby likely that the ground truth (K-L grades) produced by human experts are inconsistent which makes it impossible for ML model to learn the mapping from X-rays to grading. 
  • Are the current methods robust enough and able to generalize to data sets hiding the aforementioned pitfalls. I consider these questions as critical. I hope the authors consider these points and makes additions to the text if it feel reasonable.

Chapter 7. 

  • Please consider are the current testing processes sufficient. The authors should always report both training and test performance of the models. Low training and high test accuracy indicates that the same test set has been used too many times. In case of small data samples, in particular, it should also be confirmed that the findings are different to chance level by e.g. permutation tests. 

Chapter 7

  • I ask the author to refine the conclusions and answer to the main questions of the manuscript: what are the most important applications available in healthcare and in laboratory medicine in particular nowadays, are they reliable, safe and applied correctly. Moreover, what are the most important challenges and pitfalls of applying AI algorithms in health care and lab medicine and how they could/should be reacted.

Minor comment:

  • Please, clarify all the abbreviations (PPC, NPV, LIS, ...)

Author Response

Comments and Suggestions for Authors

The aim of the manuscript is provide an narrative review of what AI is, what application are available in healthcare and laboratory medicine. Moreover, the author's is to discuss challenges and pitfalls of AI, especially about the concern of whether the humans will be replaced by AI systems in near future.

The topic of the manuscript is important and timely due to prevailing AI hype during the recent years.  The paper also refers to some unrealized AI promises by industry that have led to both unrealistic expectations and fears in the fields of health care and medicine.

The paper is also well-organized and easy to read. Scientifically the manuscript is on a rather general level. It is clear and comprehensive, but does not provide much scientific facts concerning performance, weaknesses, challenges of AI methods for researchers. Therefore, the manuscript might be more relevant in some popular science journal, but as a narrative review can be considered relevant in the applied journal.

I am thankful for the reviewers comments as they exactly reflect the intention of this article. It should be a possibility of acquaintance to this topic for laboratory specialists who haven’t dived into the vast world of artificial intelligence yet. I held it purposely on a general level to avoid overwhelming the reader with details which might scare him/her off. I truly believe that this topic is groundbreaking for our profession and as there is no “general” overview on this topic specifically focusing on laboratory medicine, I felt the need of providing one, hopefully educating my colleagues in order to keep up with this groundbreaking development.

I present some comments that I hope the author consider before the final acceptance. My review is primarily based on the machine learning expertise with a lot of collaboration with medical and health care specialists. Therefore, my comments are mainly related to machine learning.

Chapter 2.

  • I would elaborate the description about AI and ML. For this paper it is important describe more precisely the difference between classical AI versus ML based AI that has taken the field from exhaustive human programming efforts to learning machines. The algorithms are not anymore task-specific, but the models are since the learn from the given data. It might be also good to explain most essential terms such as learning algorithms (the general algorithm that fits the models to the data), model and training, validation and testing data.
    Indeed this is a very important aspect. I have added several paragraphs as well as a figure (Figure4) to clarify these issues.

  • Furthermore, it is important to describe difference between ML and DL. The major difference between the two is the laborious feature engineering phase whose role has shrunk due to deep learning methods. Consequently sequential, spatial and temporal information existing in images, biosignals, anamnesis can be utilized in the training process more efficiently.
    I added the information in chapter 2

  • Deep learning has become popular due to the available open-source libraries, GPUs processors and adequate raw data (e.g. images)
    I added the information in chapter 2
  • Deep learning and related technological advances make things easier for data scientists, but what I always tell to medical specialists is that the modern AI models are still highly task specific. They can only what they have seen in the data. So there is really no reason to worry about machines roles.
    I agree. I elaborated on this issue in chapter 8.

  • Reconsider the sentence "Each "neuron" at the input level of such a network holds a number between 0 and 1 ...." The number depends on the type of preprocessing, it can also be less than 0 or greater 1.
    I added the word “usually" but refrained from providing additional examples to make the article as readable as possible.

  • The sentence "On the output level..." is unclear. Moreover, it does not take into account regression models in which the output(s) can be any real numbers.
    I revised the sentence.

  • Table 1 is questionable
    • I do not agree that performance of ML model does not increase with the amount of data. I think it increases for both, but due to the greater learning capacity the performance of DL models saturate later.
    • I don't understand the sentence: "Can function with a small amount of data"? 
    • ML methods may also require a lot of computational power, for example, training a large number of classical MLPs on small sample EEG data may require randomization for avoiding chance models. This may require a huge amount of computation.
    • "Tracing back the problem solving path" This is true for some traditional methods, but not necessarily for all, for instance classical MLPs, SVMs,,
    • The four statements in Table 1 should be also supported by some references.
  • I agree with the reviewer that the table indeed is debatable and quite superficiant and therefore chose to delete the table

Chapter 3

  • I think the content of this chapter is mainly irrelevant for the manuscript
    Indeed, the chapter might be irrelevant, however, I wanted to provide a smooth transition from theory to medical practical relevance by keeping the informational flow relatable, ensuring the readers attention.

Chapter 4

  • Line 276: The sentence "How efficient this device is ...." is inappropriate for a scientific paper. It sounds like marketing without any real evidence. A random observation is not evidence of anything.
    Again, this was to make the information more relatable and less abstract/theoretical. However, I see the point of the reviewer and deleted the sentence.

Chapter 5

  • Without any information about training, validation and testing process/strategy and related data sampling the performance measures are difficult to interpret, but can be sufficient for a narrative review 
    I absolutely agree with the reviewer. In this chapter however, I strictly refer to data published elsewhere.

Chapter 6

  • Data quality (noise, erroneous and incomplete data) is always an issue. Since filtering and pruning data is also a risk (of distorting data distribution),  I would suggest to discuss also about robust learning methods that tolerate noisy and even erroneous data without filtering by utilize robust statistical estimates. Moreover, most of the data are incomplete (part values are missing) for which methods based on preprocessing free strategies such available case strategy could be considered. 
    I mentioned the issue of data quality, but did not want to get into detail, as this is the topic of an entire article from A. Bietenbeck which will be published in the same special issue – in the final verion there will be an adeqquate reference. For now, I marked it as “CROSSREF”

  • Inconsistent ground truth data is also an issue. For instance, a large number of DL models have been developed to perform K-L grading for knee X-rays, but without real success in early detection. It is thereby likely that the ground truth (K-L grades) produced by human experts are inconsistent which makes it impossible for ML model to learn the mapping from X-rays to grading. 
    I absolutely agree with the reviewer and want to thank him/her for bringing this important, yet missing issue to my attention. I added a respective paragraph in chapter 6.

  • Are the current methods robust enough and able to generalize to data sets hiding the aforementioned pitfalls. I consider these questions as critical. I hope the authors consider these points and makes additions to the text if it feel reasonable.
    Agreed – I added a sentence to highlight this issue.

Chapter 7. 

  • Please consider are the current testing processes sufficient. The authors should always report both training and test performance of the models. Low training and high test accuracy indicates that the same test set has been used too many times. In case of small data samples, in particular, it should also be confirmed that the findings are different to chance level by e.g. permutation tests. 
    The mentioned articles in chapter 7 are only evaluating the outcome quality of symptom checker algorithms. However, as this issue indeed is very important, I added a paragraph in chapter 6.

  • I ask the author to refine the conclusions and answer to the main questions of the manuscript: what are the most important applications available in healthcare and in laboratory medicine in particular nowadays, are they reliable, safe and applied correctly. Moreover, what are the most important challenges and pitfalls of applying AI algorithms in health care and lab medicine and how they could/should be reacted.
    I agree and added an additional paragraph into the conclusion part.

Minor comment:

  • Please, clarify all the abbreviations (PPV, NPV, LIS, ...)
    Information added.